# Effects of Additional Dietary Fiber Supplements on Pregnant Women with Gestational Diabetes: A Systematic Review and Meta-Analysis of Randomized Controlled Studies

**DOI:** 10.3390/nu14214626

**Published:** 2022-11-02

**Authors:** Jihan Sun, Jinjing Wang, Wenqing Ma, Miao Miao, Guiju Sun

**Affiliations:** 1Key Laboratory of Environmental Medicine and Engineering of Ministry of Education, Department of Nutrition and Food Hygiene, School of Public Health, Southeast University, Nanjing 210009, China; 2Department of Nutrition, Nanjing Maternal and Child Health Hospital, Nanjing 210009, China

**Keywords:** dietary fiber, gestational diabetes, additional supplements

## Abstract

The efficacy of different types and doses of dietary fiber supplementation in the treatment of gestational diabetes (GDM) remains controversial. The purpose of this study is to investigate the effect of dietary fiber on blood glucose control in pregnant women with gestational diabetes mellitus, and further observe the effect on their blood lipids and pregnancy outcomes. We searched on Web of Science, PubMed, Embase, Scopus, and Cochrane, and included several articles on additional fortification with dietary fiber for gestational diabetes interventions. This meta-analysis included 8 trials. We found that additional dietary fiber supplements significantly reduced fasting glucose (Hedges’g = −0.3; 95% CI [−0.49, −0.1]), two-hour postprandial glucose (Hedges’g = −0.69; 95% CI [−0.88, −0.51]), glycated hemoglobin (Hedges’g = −0.5; 95% CI [−0.68, −0.31]), TC (Hedges’g = −0.44; 95% CI [−0.69, −0.19]), TG (Hedges’g = −0.3; 95% CI [−0.4, −0.2]) and LDL-C (Hedges’g = −0.48; 95% CI [−0.63, −0.33]). It also significantly reduced preterm delivery (Hedges’g = 0.4, 95% CI [0.19~0.84]), cesarean delivery (Hedges’g = 0.6; 95% CI [0.37~0.97]), fetal distress (Hedges’g = 0.51; 95% CI [0.22~1.19]), and neonatal weight (Hedges’g = −0.17; 95% CI [−0.27~−0.07]). In a subgroup analysis comparing dietary fiber type and dose, insoluble dietary fiber was more effective than soluble dietary fiber in reducing fasting glucose (Hedges’g = −0.44; 95% CI [−0.52, −0.35]). ≥12 g fiber per day may be more effective in improving glycemic lipid and pregnancy outcomes than <12 g/day, but the difference was not statistically significant. In conclusion, our meta-analysis showed that dietary fiber supplementation significantly improved glycolipid metabolism and pregnancy outcomes in gestational diabetes. Dietary fiber may be considered adjunctive therapy for gestational diabetes, and an additional supplement with insoluble dietary fiber is more recommended for those with poor fasting glucose. However, more high-quality studies are needed on the further effect of fiber type and the dose-effect relationship.

## 1. Introduction

Gestational diabetes mellitus (GDM) is usually defined as the first episode or finding of carbohydrate intolerance during pregnancy [1]. GDM increases the risk of maternal cardiovascular disease and type 2 diabetes in later life, the incidence of large births, and neonatal complications. There is also a long-term risk of obesity, type 2 diabetes, and cardiovascular disease in later offspring [2]. In a recent Japanese birth cohort study, it was found that mothers with gestational diabetes had a lower average gestational age, significantly heavier placental weight, and a higher relative risk of delivery complications and neonatal complications compared to normal pregnant women [3]. Results of a cross-sectional study in Thailand from September 2018 to February 2019 showed an overall prevalence of gestational diabetes of 18.6%, significantly lower pregnancy weight gain, and a higher prevalence of pre-eclampsia and macrosomia in pregnant women with gestational diabetes compared to those without gestational diabetes [4]. Gestational diabetes was positively associated with the risk of offspring acute lymphoblastic leukemia (OR = 1.40, 95% CI = 1.12 to 1.75; *I*^2^ = 0.0%) [5].

Treatment for gestational diabetes includes diet, lifestyle, and medication. Medication such as metformin, glyburide, or insulin is recommended only when diet and lifestyle changes are not effective in controlling blood glucose levels [6]. Nutrition has an important role in the risk of GDM, and nutritional supplements may be a safe and effective means of treating GDM, such as Inositol, Vitamins, Minerals, Fatty Acids, Probiotics, and Fiber [7]. Fiber is part of a healthy diet for diabetes treatment. A meta-analysis of diabetes showed that a high-fiber diet is an important component of diabetes management, improving glycemic control, lipids, body weight, and inflammatory markers, and reducing premature mortality, with significant effects for any fiber type, any dose, or any type of diabetes, but there may be a dose-effect relationship, and an increase in fiber intake of 15 or 35 g per day may be a reasonable goal [8]. The prevalence of GDM in China is 17.5%, and an analysis of 9317 women found that women with the highest pre-pregnancy dietary fiber intake had a significantly lower risk of developing gestational diabetes mellitus. In addition, increased GI or GL and decreased fiber intake during gestation were independently associated with poor development of fasting glucose, glycated hemoglobin, and insulin resistance [9]. The intake of dietary fiber in various types of foods during mid-pregnancy may be associated with the risk of GDM. In particular, a diet rich in total fiber and fruit fiber may help improve blood glucose [10]. In a randomized controlled trial, dietary blueberry and soluble fiber supplements reduced the risk of GDM in obese women [11].

The preventive and ameliorative effects of dietary fiber on GDM have been well documented in many studies, but there are few studies on the effects of additional dietary fiber supplements and different fiber types and doses on GDM, for this reason, this study conducted a meta-analysis from a systematic search of randomized controlled trials to assess the effects of fiber fortification on indicators of glycemic control, lipids, pregnancy outcome, and neonatal outcome in GDM, and further subgroup analyses were conducted to investigate the differences in dietary fiber type and amount of fortification on these outcomes.

## 2. Methods

This systematic review and meta-analyses were conducted following Cochrane’s PRISMA (Preferred Reporting Items for Systematic Reviews and Meta-Analyses) guidelines, registration number CRD42022363892.

### 2.1. Search Strategy

The articles were searched in five databases: Pubmed, Embase, Scopus, Cochrane, and Web of Science. The search string included (“Dietary Fiber” OR “Dietary Fibers” OR “Fibers, Dietary” OR “Fiber, Dietary” OR “Wheat Bran” OR “Bran, Wheat” OR “Brans, Wheat” OR “Wheat Brans” OR “Roughage” OR “Roughages” OR “inulin” OR “inuline” OR “pectin” OR “beta glucan” OR “fructose oligosaccharide” OR “oligofructose”) AND (“Diabetes, Gestational” OR “Diabetes, Pregnancy-Induced” OR “Diabetes, Pregnancy Induced” OR “Pregnancy-Induced Diabetes” OR “Gestational Diabetes” OR “Diabetes Mellitus, Gestational” OR “Gestational Diabetes Mellitus”).

### 2.2. Inclusion and Exclusion Criteria

Literature was included if it met the following criteria: (1) Study was a randomized controlled trial, (2) At least one outcome of interest was reported—Fasting plasma glucose, Blood glucose two hours after a meal, (3) The intervention included only dietary fiber products compared with the control group and, (4) The intervention objects were pregnant women with gestational diabetes mellitus. Exclusion criteria were as follow: (1) There were other interventions besides fiber fortification and (2) The experiment was not designed for eligible human subjects. When there was a difference in literature screening, the authors (J.H.S, J.J.W and W.Q.M) discussed and solved it.

### 2.3. Data Extraction and Quality Assessment

Two independent reviewers extracted the following basic information from selected articles: author, year, sample size, blinding method, duration of intervention, type of fiber, intervention dose, and changes in key indicators before and after the intervention. Two articles were excluded because lacking standard deviation and the authors contacted by email did not respond.

The quality of the included literature was evaluated using an improved Cochrane bias risk assessment tool. Assessing the risk of bias: (1) random sequence generation; (2) allocation concealment; (3) blinding of participants and personnel; (4) blinding of outcome assessment; (5) incomplete outcome data; (6) selective outcome reporting; (7) other bias. Three levels were described for each item: “high risk”, “low risk”, and “unclear”.

### 2.4. Statistical Analysis

RevMan 5.4 was used for quality assessment provided by Cochrane collaborate and Standard errors (SE) reported in articles were converted to SD. Units for TG, TC, HDL-C, LDL-C, and blood glucose concentrations were standardized to mmol/L. Stata 17 was used for the statistical analysis of the extracted data.

*p* < 0.05 was considered statistically significant. Heterogeneity was assessed by the *I*^2^ index, *I*^2^ is the portion(%) of the total variability attributed to pure heterogeneity among studies. When *I*^2^ is 0, it means that studies are completely homogeneous. If *I*^2^ > 50%, it indicates there is heterogeneity in studies. We used the random effects model for analysis. Estimates were statistically different (*p* < 0.05; both overall effect sizes fell outside the 95% CI of the counterpart) and the boundaries of the 95% CI had the same sign, bias can be considered influential.

Subgroup analysis was performed by fiber type (soluble fiber, insoluble fiber, and complex) and the amount of fortified fiber per day (<12 g/day vs. ≥12 g/day). The 12 g quantity of fiber was determined by the difference between the average dietary intake of 13 g of fiber [9] and the recommended intake of 25 g to 30 g for women in the second trimester of pregnancy.

## 3. Results

### 3.1. Search Results and Characteristics

Figure 1 shows a flow chart from search to meta-analysis. A total of 614 articles were obtained, and after duplicate removal of 243 articles, 371 articles remained for screening. 361 articles were excluded based on the criteria. Two articles of the remaining 10 articles [12,13] were excluded, one because a loading meal method was used and the other because the authors could not be contacted to obtain useful data, and a total of eight articles were finally used for the systematic review and meta-analysis.

Table 1 summarized the characteristics of the study, which included eight articles with durations ranging from 2 to 12 weeks. There were three articles with <12 g/day fiber fortification and five articles with ≥12 g/day fiber fortification. Three articles used soluble fiber-fortified foods, three articles used insoluble fiber-fortified foods, and two articles used complex fiber. Each randomized controlled trial had a corresponding control diet or non-fortified placebo. The results obtained were: fasting glucose, two-hour postprandial glucose, glycated hemoglobin, triglycerides, cholesterol, HDL, LDL, and pregnancy outcome (preterm delivery, cesarean delivery, fetal distress, and neonatal weight).

### 3.2. Quality and Risk of Bias within Studies

The risk of bias assessment is shown in Figure 2. In terms of the random sequence generation method, five had sufficient random components with a low risk of bias, two were determined to be unclear, and one was high risk. For allocation hiding, two articles performed low risk and six were unclear. Most randomized controlled trials were single-blinded, and three articles were unclearly blinded. Six articles indicated blinding of outcome assessors, and two were unclear.

### 3.3. Meta-Analyses

The overall meta-analysis effect sizes and confidence intervals for each outcome are summarized in Table 2. Overall, there was a high degree of heterogeneity in fasting glucose and triglycerides.

### 3.4. Serum Glucose Outcomes

The Figure 3 showed a significant decrease in fasting glucose (Hedges’g = −0.3; 95% CI [−0.49, −0.1]; *I*^2^ = 83%; 8 articles) and two-hour plasma glucose (Hedges’g = −0.69; 95% CI [−0.88, −0.51]; *I*^2^ = 49%; 7 articles), glycated hemoglobin (Hedges’g = −0.5; 95% CI [−0.68, −0.31]; *I*^2^ = 0%; 2 articles), and number of qualified blood glucose increased significantly (Hedges’g = 5.27; 95% CI [2.56, 10.83]; *I*^2^ = 0%; 2 articles).

### 3.5. Serum Lipid Outcome

The Figure 4 showed an improvement in lipids after fiber food supplement, with a decrease in TC (Hedges’g = −0.44; 95% CI [−0.69, −0.19]; *I*^2^ = 56%; 4 articles), TG (Hedges’g = −0.3; 95% CI [−0.4, −0.2]; *I*^2^ = 0%; 4 articles), decreased LDL-C (Hedges’g = −0.48; 95% CI [−0.63, −0.33]; *I*^2^ = 0%; 2 articles), however HDL-C was not significantly different (Hedges’g = 0.03; 95% CI [ −0.06, 0.11]; *I*^2^ = 0%; 2 articles).

### 3.6. Pregnancy and Neonatal Outcomes

As shown in Figure 5, compared to placebo or control, there were significantly fewer preterm deliveries (Hedges’g = 0.4, 95% CI [0.19, 0.84]; *I*^2^ = 0%; 3 articles), significantly fewer cesarean deliveries (Hedges’g = 0.6; 95% CI [0.37~0.97]; *I*^2^ = 0%; 3 articles), and significantly fewer fetal distress (Hedges’g = 0.51; 95% CI [0.22~1.19]; *I*^2^ = 0%; 2 articles) and a significant reduction in neonatal weight (Hedges’g = −0.17; 95% CI [−0.27~−0.07]; *I*^2^ = 0%; 2 articles).

### 3.7. Subgroup Analyses on Fiber Type, Fiber Quantity

Table 3 shows the results of the subgroup analysis. HbA1c, TC, TG, HDL, LDL, and pregnancy outcomes were not analyzed in subgroups of fiber type, because in at least one group there was only one comparison. Subgroup analysis for HDL, LDL, and pregnancy outcomes on fiber quantity was not conducted for the same reason.

### 3.8. Fiber Type

The intervention effect of insoluble dietary fiber on fasting glucose was reduced (Hedges’g = −0.44; 95% CI [−0.52, −0.35]; *I*^2^ = 33.8%) but not for soluble and complex fiber. 2-h glucose was not affected by fiber type (Figure 6).

### 3.9. Fiber Quantity

The ≥12 g group significantly reduced fasting glucose (Hedges’g = −0.40; 95% CI [−0.69, −0.11]; *I*^2^ = 87%), but two-hour postprandial glucose (Hedges’g = −0.84; 95% CI [−1.22, −0.46]; *I*^2^ = 51%), TC (Hedges’g = −0.62; 95% CI [−0.87, −0.36]; *I*^2^ = 20%), and TG (Hedges’g = 0.34; 95% CI [−0.58, −0.09]; *I*^2^ = 0%) were not statistically significant (Figure 7 and Figure 8). In addition, Figure 9 shows that there was no significant difference between the different doses in pregnancy outcomes and neonatal outcomes.

### 3.10. Sensitivity Analyses and Publication Bias

Funnel plots (Figure 10) were used to qualitatively evaluate publication bias and Egger’s test was used to quantitatively determine publication bias. The results showed no significant publication bias for fasting glucose (t = −1.11, *p* = 0.311) and two-hour postprandial glucose (t = 0.45, *p* = 0.671).

## 4. Discussion

In general, dietary fiber is the edible part of plants, or similar carbohydrates, which resist digestion and absorption in the intestine. Dietary fiber can be divided into many different fractions, including arabinoxylan, inulin, pectin, bran, cellulose, beta-glucan, and resistant starch. The mechanisms of the metabolic health effects of dietary fiber may be related to changes in intestinal viscosity, nutrient absorption, rate of transmission, short-chain fatty acid production, and intestinal hormone production [22]. Chinese women are part of the international high-risk group for gestational diabetes mellitus (GDM), and a meta-analysis of Chinese women with GDM showed that a low-GI diet, a low-GL diet, and a fiber-rich diet were associated with improved glycemic control and pregnancy outcomes [23]. A dietary pattern with more rice, beans, and vegetables and fewer full-fat dairy products, cookies, and sweets had a higher fiber density and were negatively associated with thrombosis index and gestational diabetes [24].

### 4.1. Effects of Fiber-Fortified Food on Serum Glucose Outcomes

Dietary fiber plays an important role in the control of postprandial glucose and insulin response in diabetic patients. Dietary fiber intake has been shown to slow gastric emptying in healthy subjects, and similarly, the effect of soluble dietary fiber in improving postprandial glucose in patients with type 2 diabetes is associated with slower gastric emptying [25]. During a short-term intervention, increased intake of soluble dietary fiber significantly improved blood glucose levels, insulin resistance, and metabolic profiles in diabetic patients, but not islet secretion [26]. Insoluble oat fiber can also effectively affect blood glucose metabolism, with the most pronounced effect in subjects with impaired fasting glucose, even alone [27]. Different types of dietary fiber may differ in carbohydrate uptake and metabolism [28]. In China, results of a prospective analysis of the association between dietary fiber intake and the risk of developing prediabetes in Chinese adults showed that fiber from fruits, but not from grains, legumes, and vegetables, was negatively associated with prediabetes. Intake of total dietary fiber, soluble fiber, and fiber from fruits was associated with a lower risk of prediabetes [29]. The traditional view is that viscosity and solubility are the main reasons why dietary fiber improves blood glucose. However, a study used enzymatic extraction of barley insoluble fiber (BIF) and soluble fiber (BSF) to compare the anti-diabetic effects and found that both had hypoglycemic lipidemic effects but may act through different mechanisms in the intestinal flora [30]. The results of our meta-analysis showed that additional dietary fiber fortification significantly improved fasting and postprandial glucose and glycated hemoglobin in patients with gestational diabetes, and that insoluble dietary fiber may be more effective in improving fasting glucose, similar to the results of Kabisch’s study [27], suggesting that there are other reasons beyond viscosity and solubility that influence the glycemic improvement of different types of dietary fiber. Consuming foods high in dietary fiber may help prevent diabetes. Studies have shown that in the general Japanese population, a higher intake of dietary fiber is associated with a lower risk of developing type 2 diabetes [31]. A prospective cohort study of dietary fiber intake and risk of Type 2 Diabetes showed a non-linear relationship between total dietary fiber intake and risk of Type 2 Diabetes [32]. The results of this meta-analysis showed that dietary fiber fortification with ≥12 g/day was more effective in improving fasting blood glucose than <12 g/day, and there was no significant difference in two-hour postprandial blood glucose.

### 4.2. Effects of Fiber-Fortified Food on Lipid Metabolism

The role of dietary fiber in regulating lipid metabolism has been confirmed by numerous studies. Consumption of soluble fiber can reduce cholesterol and LDL levels by about 5–10%, but changes in HDL or triglyceride levels are minimal, and high molecular weight fiber is more effective in reducing lipid levels [33]. The lipid-lowering effect of fiber may be related to its viscosity, the higher the viscosity of fiber, the better the lipid-lowering effect [34]. However, the results of a study evaluating the relationship between dietary fiber sources and cardiovascular risk factors in a Spanish population showed that higher insoluble fiber intake has an important role in the control and management of hypertension, blood lipids, and methionine [35]. Since there was only one article in at least one subgroup in this study, the effect of fiber type on lipids was not included in the analysis, and more studies are needed to discuss it. Dietary fiber intake also has an effect on blood lipids. In diabetic patients, a decrease in dietary fiber intake was positively associated with cholesterol and LDL levels [36]. In addition, pre-pregnancy dietary patterns were also associated with gestational lipid levels, with higher fast food and candy pattern scores associated with higher triglyceride levels and slower HDL-C changes during pregnancy, while higher vegetable and dairy dietary pattern scores were associated with faster HDL-C changes during pregnancy [37]. The results of this study showed that higher fiber intake may be more effective in lowering blood lipids, but the difference between the two groups was not statistically significant, and it is possible that the additional dietary fiber fortification in both groups far exceeded the recommended intake, and confounding factors such as diet need to be excluded.

### 4.3. Effects of Fiber-Fortified Food on Pregnancy and Neonatal Outcome

Dietary interventions in early pregnancy have a positive impact on maternal gastrointestinal index, nutrient intake, and weight gain during pregnancy. Increased dietary fiber intake may prevent excessive maternal weight gain and reduce infant birth weight [38]. Pre-pregnancy dietary patterns were also associated with neonatal outcomes, with a higher intake of fast food and sweets increasing the rate of large births, while vegetable and dairy dietary patterns reduced the chances of preterm birth [39]. Processed diets high in fat and low in fiber reduce gut microbiota alpha diversity thus affecting spontaneous preterm birth (SPTB) [40]. Our study showed similar results. Dietary fiber reduced the incidence of preterm delivery, cesarean delivery, and fetal distress, and also significantly decreased neonatal weight. Higher fiber intake may have a better effect on pregnancy and neonatal outcomes, but the difference between the two groups was not statistically significant. There was only one article on at least one subgroup type, for this reason, the effect of fiber type on blood lipids was not included in the analysis, and more studies are needed to discuss it.

### 4.4. Strengths, Limitations, and Insights

This article reviewed the beneficial improvements of fiber fortification in pregnant women with gestational diabetes, not just fiber in the general diet. The different benefits of different fiber types and doses were also highlighted. This provides valuable insights into the management of glycemic control in pregnant women.

Due to the low number of articles included in this meta-analysis, the publication bias derived from the funnel plot may not be conclusive, and more experiments are needed to further prove our point in the future.

## 5. Conclusions

In conclusion, evidence from our meta-analysis suggested that additional dietary fiber supplementation significantly reduced fasting and two-hour postprandial glucose in people with gestational diabetes. In addition, it also assisted in reducing lipid levels and improving adverse pregnancy and neonatal outcomes. However, there are some limitations in this review, so further high-quality and large-sample-size studies are needed to validate the effects of different fiber types and doses on outcomes.

## Figures and Tables

**Figure 1 nutrients-14-04626-f001:**
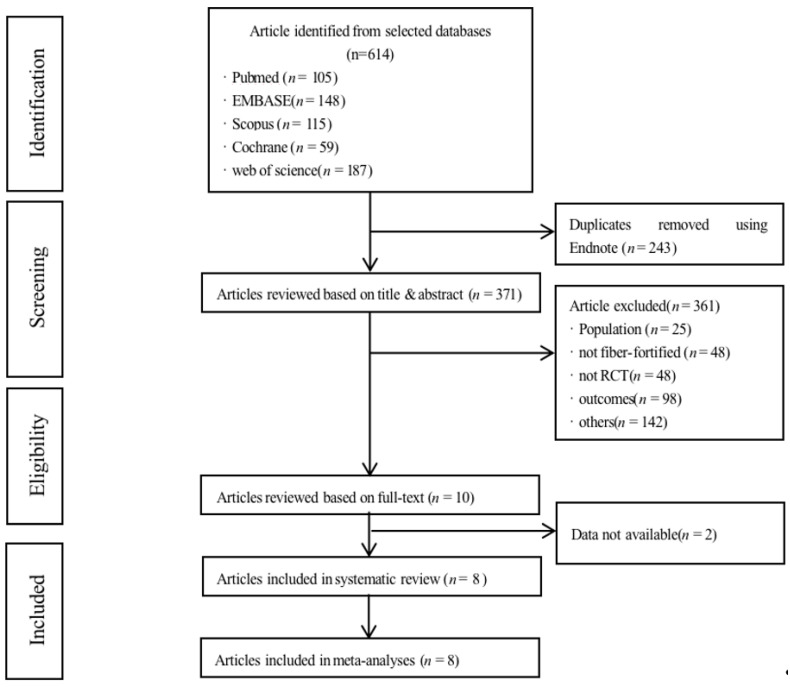
Flow chart from search to meta-analysis.

**Figure 2 nutrients-14-04626-f002:**
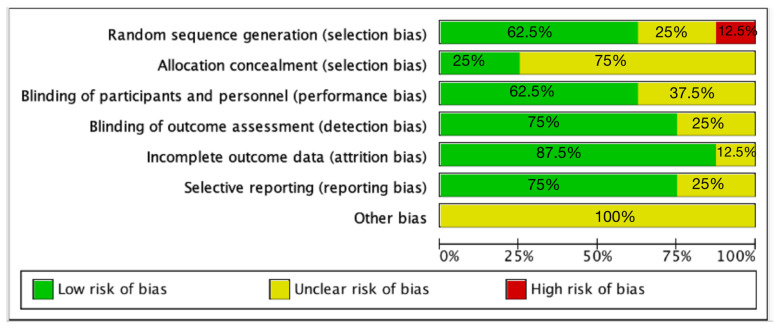
The risk of bias assessment.

**Figure 3 nutrients-14-04626-f003:**
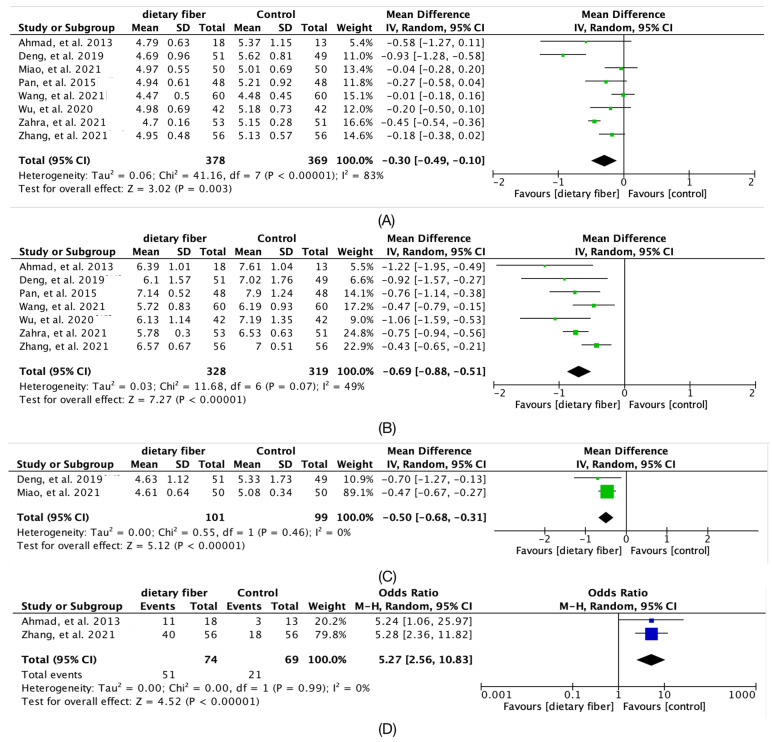
Forest plot for the overall meta-analysis of Serum glucose outcomes: (**A**) fasting glucose (mmol/L); (**B**) 2 h plasma glucose (mmol/L); (**C**) HbA1c (mmol/L); (**D**) Number of qualified blood glucose [14,15,16,17,18,19,20,21].

**Figure 4 nutrients-14-04626-f004:**
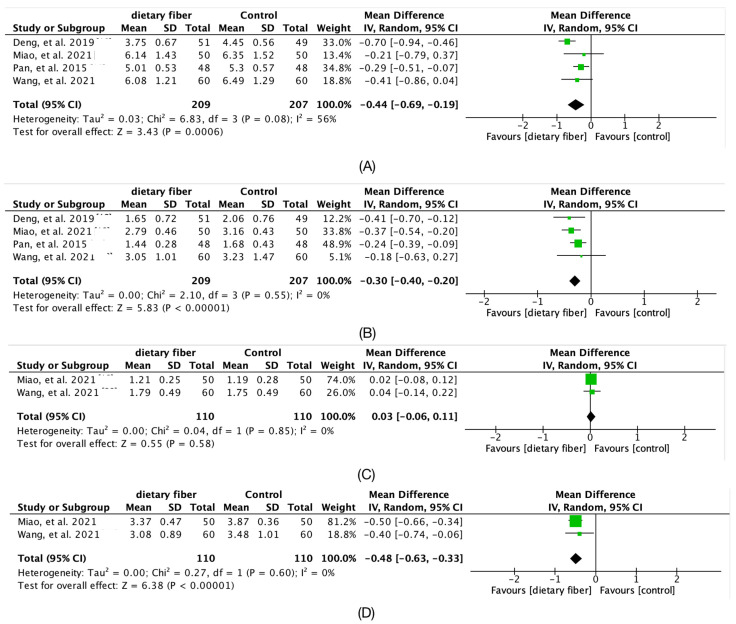
Forest plot for the overall meta-analysis of Serum lipid outcome: (**A**) Serum total cholesterol (TC, mmol/L); (**B**) Triglyceride (TG, mmol/L); (**C**) High-density lipoprotein cholesterol (HDL, mmol/L); (**D**) Low-density lipoprotein cholesterol (LDL, mmol/L) [15,16,18,19,20,21].

**Figure 5 nutrients-14-04626-f005:**
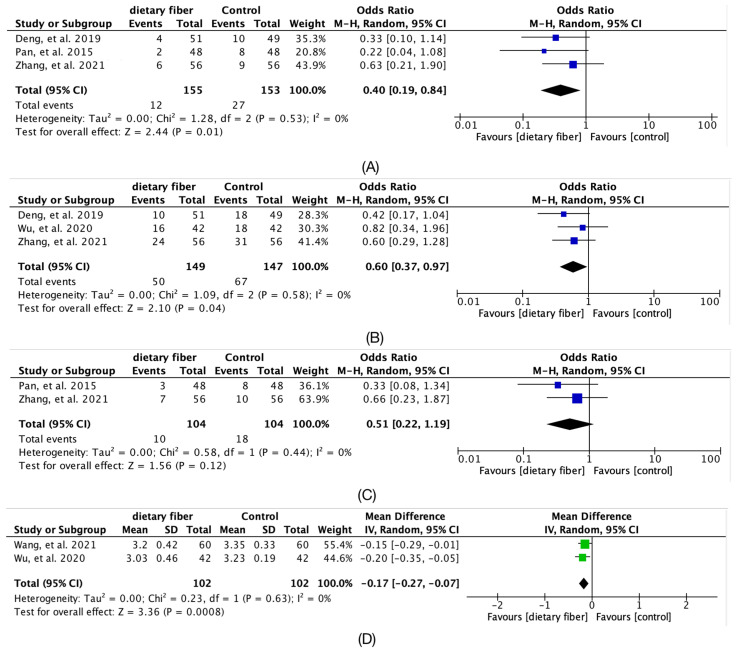
Forest plot for the overall meta-analysis of Pregnancy and neonatal outcomes: (**A**) preterm deliveries; (**B**) cesarean deliveries; (**C**) fetal distress; (**D**) neonatal weight [15,16,17,21].

**Figure 6 nutrients-14-04626-f006:**
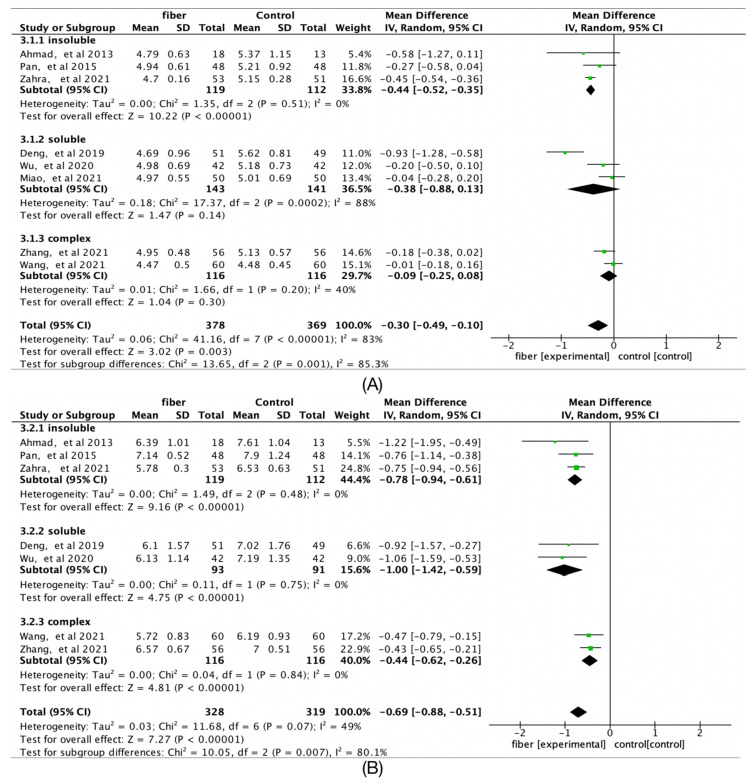
Forest plots of subgroup analysis of fiber type on (**A**) fasting glucose (mmol/L) and (**B**) 2-h glucose (mmol/L) [14,15,16,17,18,19,20,21].

**Figure 7 nutrients-14-04626-f007:**
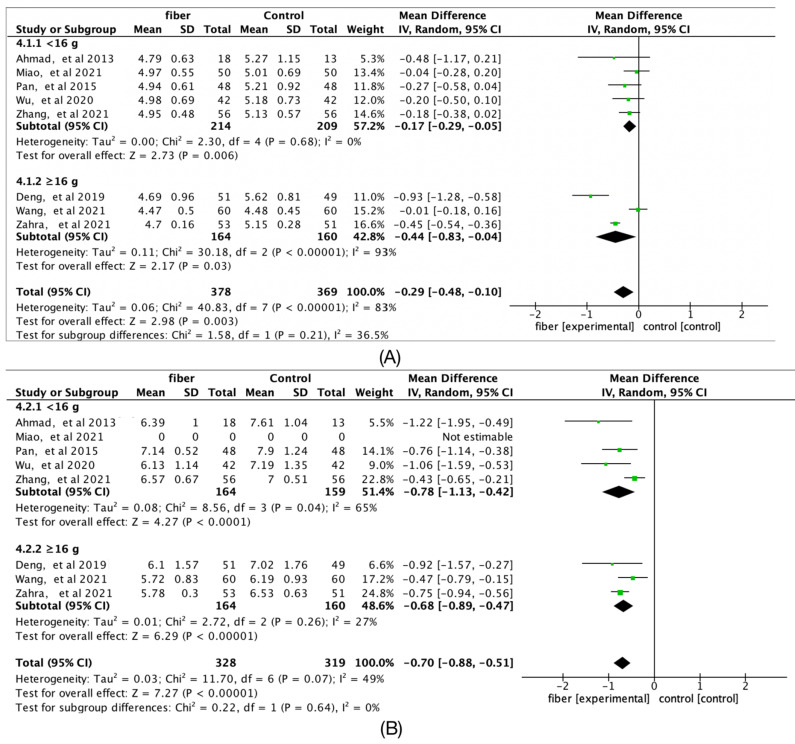
Forest plots of subgroup analysis of fiber quantity on (**A**) fasting glucose (mmol/L) and (**B**) 2-h glucose (mmol/L) [14,15,16,17,18,19,20,21].

**Figure 8 nutrients-14-04626-f008:**
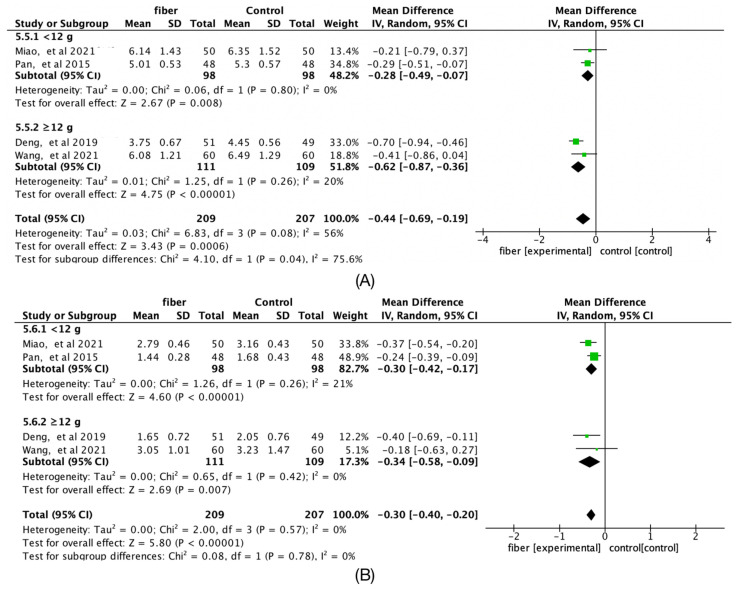
Forest plots of subgroup analysis of fiber quantity on (**A**) Total cholesterol (TC, mmol/L) and (**B**) Triglycerides (TG, mmol/L) [15,16,18,20].

**Figure 9 nutrients-14-04626-f009:**
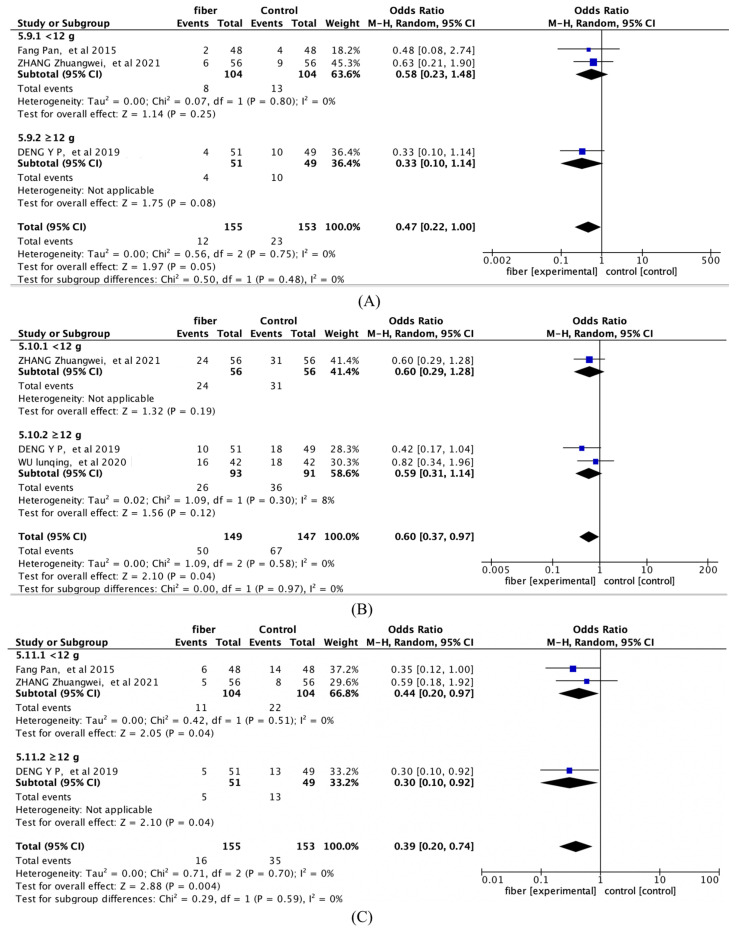
Forest plots of subgroup analysis of fiber quantity on (**A**) preterm deliveries, (**B**) cesarean deliveries, and (**C**) macrosomia [15,16,17,21].

**Figure 10 nutrients-14-04626-f010:**
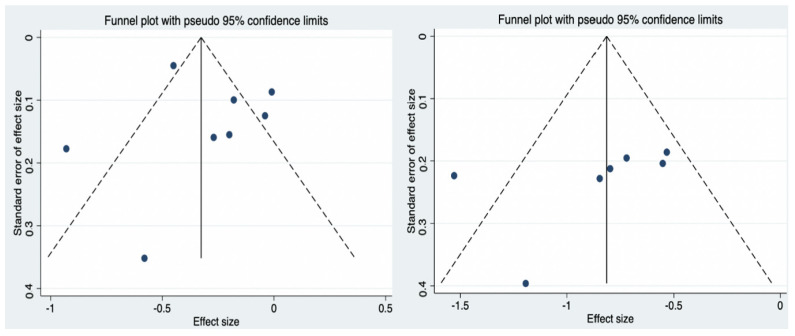
Funnel plot of publication bias for the primary outcomes.

**Table 1 nutrients-14-04626-t001:** Study characteristics of the eight included articles.

NO	First Author, Year	Population Size and Description	Intervention Duration (Weeks)	Study Design	Control Food and Description	Fiber and Description
1	Ahmad, et al., 2013 [14]	36 subjects with GDM(18 I; 13 C)	2	Parallel Single-blind	low GL diet	15 g insoluble fiber (wheat bran)
2	Pan, et al., 2015 [15]	96 subjects with GDM (48 I; 48 C)	4	Parallel Single-blind	Diet therapy	10.5 g insoluble fiber (wheat fiber)
3	Deng, et al., 2019 [16]	100 subjects with GDM (51 I; 49 C)	12	Parallel Single-blind	Basic dietary nutrition support treatment	20 g soluble dietary fiber (Fiber polysaccharide)
4	Wu, et al., 2020 [17]	84 subjects with GDM (42 I; 42 C)	8	Parallel Single-blind	Personalized diet control	15 g soluble dietary fiber (Inulin, stachyose, microcrystalline cellulose, oat fiber)
5	Miao, et al., 2021 [18]	100 subjects with GDM (50 I; 50 C)	4	Parallel Single-blind	Dietary guidelines	10 g soluble dietary fiber (Inulin)
6	Zahra, et al., 2021 [19]	104 subjects with GDM (53 I; 51 C)	4	Parallel Single-blind	Dietary guidelines	30 g insoluble fiber (oat bran)
7	Wang, et al., 2021 [20]	120 subjects with GDM (60 I; 60 C)	8	Parallel Single-blind	Dietary guidelines	19 g complex dietary fiber (Ricnoat)
8	Zhang, et al., 2021 [21]	112 subjects with GDM (56 I; 56 C)	8	Parallel Single-blind	Dietary guidelines	9.5 g complex dietary fiber (Ricnoat)

Gestational diabetes mellitus (GDM); Glycemic Index (GI); Intervention (I); Control (C).

**Table 2 nutrients-14-04626-t002:** Effect sizes and confidence intervals for each outcome.

Outcome	Hedges’ g [95% CI]	*I*^2^ Value (%)	*p* Value
fasting glucose	−0.3 [−0.49, −0.1]	83	0.003
Two-hour plasma glucose	−0.69 [−0.88, −0.51]	49	<0.001
HbA1c	−0.5 [−0.68, −0.31]	0	<0.001
Number of qualified blood glucose	5.27 [2.56, 10.83]	0	<0.001
TC	−0.44 [−0.69, −0.19]	56	<0.001
TG	−0.3 [−0.4, −0.2]	0	<0.001
HDL	−0.03 [−0.06, 0.11]	0	0.58
LDL	−0.48 [−0.63, −0.33]	0	<0.001
preterm delivery	0.4 [0.19, 0.84]	0	0.01
cesarean delivery	0.6 [0.37, 0.97]	0	0.04
fetal distress	0.51 [0.22, 1.19]	0	0.12
neonatal weight	−0.17 [−0.27, −0.07]	0	<0.001

Glycated hemoglobin (HbA1c); Serum total cholesterol (TC); Triglyceride (TG); High-density lipoprotein cholesterol (HDL); Low-density lipoprotein cholesterol (LDL).

**Table 3 nutrients-14-04626-t003:** The results of subgroup analysis.

Outcome	Fiber Type						Fiber Quantity			
	Insoluble		Soluble		Complex		<12 g		≥12 g	
	Hedges’g (95% CI)	*I*^2^(%)	Hedges’g (95% CI)	*I*^2^(%)	Hedges’g (95% CI)	*I*^2^(%)	Hedges’g (95% CI)	*I*^2^(%)	Hedges’g (95% CI)	*I*^2^(%)
fasting glucose	−0.44 [−0.52, −0.35]	33.8	−0.38 [−0.88, 0.13]	36.5	−0.09 [−0.25, 0.08]	29.7	−0.15 [−0.29, −0.02]	0	−0.40 [−0.69, −0.11]	87
2-h glucose	−0.77 [−0.94, −0.61]	44.4	−1.00 [−1.42, −0.59]	15.5	−0.44 [−0.62, −0.26]	40.1	−0.56 [−0.87, −0.24]	54	−0.84 [−1.22, −0.46]	51
TC	—	—	—	—	—	—	−0.28 [−0.49, −0.07]	0	−0.62 [−0.87, −0.36]	20
TG	—	—	—	—	—	—	−0.30 [−0.42, −0.17]	21	−0.34 [−0.58, −0.09]	0

Serum total cholesterol (TC); Triglyceride (TG).

## Data Availability

Data is contained within the article.

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
