# Peer review of "Effects of Additional Dietary Fiber Supplements on Pregnant Women with Gestational Diabetes: A Systematic Review and Meta-Analysis of Randomized Controlled Studies"

_nutrients, 2022, doi:10.3390/nu14214626_

Round 1

Reviewer 1 Report

This article used the meta-analytic review of randomized controlled trials to summarize the evidence on the effects of dietary fiber supplementation on various indicators of glucose and lipid metabolisms in pregnant women with gestational diabetes mellitus.Studies on the positive effects of high-fiber diet on blood glucose are common. This study further re-examined the interventional effects of additional fiber consumption with different dosages and fiber types in gestational diabetes mellitus. The topic is novel and timing, and I think it is very insightful. However, the article has some minor issues that need to be discussed with the authors.

Methods:

1. Subgroup analysis: the amount of fiber 12g, please explain in detail the reason for grouping.

2. How to deal with confounding factors in the meta-analysis, such as how to eliminate the different effects of dietary fiber content in their daily diet.

3. The timing of the intervention should also be included in the subgroup analysis, please add or explain why it was not explored.

4. How did you extract the data from each eligible trial? Whether the mean change with SD from baseline to the end of intervention in the indicators were eligible for your meta-analyses? Please write the statistical details clearly in Method.

5. In the section of inclusion and exclusion criteria, please add the study population’ descriptions. The study subjects were the women with gestational diabetes mellitus?

6. Most of glucose-lipid indicators were analyzed in the present study. Please write clearly which one is primary outcome and other one is secondary outcome. Maybe glucose levels are primary measurements, while blood lipids and others were secondary outcomes?

Result:

Some indicators cannot be included in the analysis due to the small number of articles, but they can be appropriately added to the discussion and some suggestions can be made.

References:

Authors should complete the information in the references.

English editing:

Several rough and poor expressions in the manuscript need to be improved, especially in abstract. For example, the tedious expression in abstract, "The aim of this study was to update and determine the hypoglycemic effect of dietary fiber and the effect of different types and doses.”“…insoluble dietary fiber was more effective in improving fasting glucose…”(incomplete expressions).

Author Response

This article used the meta-analytic review of randomized controlled trials to summarize the evidence on the effects of dietary fiber supplementation on various indicators of glucose and lipid metabolisms in pregnant women with gestational diabetes mellitus.Studies on the positive effects of high-fiber diet on blood glucose are common. This study further re-examined the interventional effects of additional fiber consumption with different dosages and fiber types in gestational diabetes mellitus. The topic is novel and timing, and I think it is very insightful. However, the article has some minor issues that need to be discussed with the authors.

Methods:

  1. Subgroup analysis: the amount of fiber 12g, please explain in detail the reason for grouping.

R1: The 12 g quantity of fiber was determined by the difference between the average dietary intake of 13g fiber9 and the recommended intake of 25g to 30g for women in the second trimester of pregnancy. (Line 121)

  1. How to deal with confounding factors in the meta-analysis, such as how to eliminate the different effects of dietary fiber content in their daily diet.

R: The control groups in the included articles were standard dietary controls, guided according to the recommendations of the dietary guidelines, which are shown in Table 1.

  1. The timing of the intervention should also be included in the subgroup analysis, please add or explain why it was not explored.

R:  In fact, we found that the effect of intervention time on blood glucose was not obvious due to the low number of articles that could be included. If more similar articles are updated in the future, we can continue to discuss the subgroups of intervention time.

  1. How did you extract the data from each eligible trial? Whether the mean change with SD from baseline to the end of intervention in the indicators were eligible for your meta-analyses? Please write the statistical details clearly in Method.

R: Standard errors (SE) reported in articles were converted to SD. Units for TG, TC, HDL-C, LDL-C, and blood glucose concentrations were standardized to mmol/L. Stata 17 were used for the statistical analysis of the extracted data.(Line 115)

  1. In the section of inclusion and exclusion criteria, please add the study population’ descriptions. The study subjects were the women with gestational diabetes mellitus?

R: The intervention subjects were added to the inclusion criteria

  1. Most of glucose-lipid indicators were analyzed in the present study. Please write clearly which one is primary outcome and other one is secondary outcome. Maybe glucose levels are primary measurements, while blood lipids and others were secondary outcomes?

R: Our main outcome measure was blood glucose, and then we discussed the effects on maternal blood lipids and pregnancy outcomes.

Result:

Some indicators cannot be included in the analysis due to the small number of articles, but they can be appropriately added to the discussion and some suggestions can be made.

R: The limitation has been added under the discussion. (line 415)

References:

Authors should complete the information in the references.

R: The reference citations have been revised

English editing:

Several rough and poor expressions in the manuscript need to be improved, especially in abstract. For example, the tedious expression in abstract, "The aim of this study was to update and determine the hypoglycemic effect of dietary fiber and the effect of different types and doses.”“…insoluble dietary fiber was more effective in improving fasting glucose…”(incomplete expressions).

R: Some of the unclear expressions have been revised, especially in the aim and results description section of the abstract.

Reviewer 2 Report

The authors have conducted an interesting research on the effects of dietary fiber on gestational diabetes. Please consider the following comments:

Comment 1: When providing the itemized response to these comments, please ensure the number of the lines where the modifications are located in the SUBMITTED REVISED version of your manuscript. Otherwise, the responses may not be assessed. Thank you.

Comment 2: Line 22 - Please clarify what "improving" means.

Comment 3: Authors should explicitly indicate if the meta-analysis is a new one (no other meta-analysis is available on the same topic) or if it is an update. If this is an update, the Discussion should be elaborated accordingly.   

Comment 4: Please ensure that all materials and methods section is written in the past tense.

Comment 5: Please clarify criterion #2 in lines 86-87.

Comment 6: Please clarify what "people" means in line 90 (human subjects?)

Comment 7: Stating exclusion criterion #3 is redundant with the inclusion criteria and should be removed.

Comment 8: Line 94 should say "blinding method."

Comment 9: In lines 94-95, please clarify what "intervention METHOD" and "TYPE" means.

Comment 10: Line 96. "two were excluded" Two what? Studies?

Comment 11: In line 97, please clarify what "incomplete" means.

Comment 12: In  line 97, should it be "by email did not respond."

Comment 13: Please clearly detail the procedure in lines 98-101.

Comment 14: This is a major observation: In lines 104-105, the value of I2 is used to indicate the significance of the heterogeneity. The terminology "Significance" should be reserved for inferences based on statistical analysis (such as in lines 103-104; P<0.05). However, in lines 104-105, "significance" is used for something different. Please rewrite this sentence and correct it where appropriate across the whole manuscript. Please consider that a meta-analysis with a low I2 could have only trivial heterogeneity but could also have substantial heterogeneity. Conversely, a meta-analysis with a high I2 could have substantial heterogeneity, but could also have only trivial heterogeneity. A concise definition recently used in a partner journal of MDPI is the following: "I2 is defined as the portion (%) of the total variability attributed to pure heterogeneity among studies" (https://www.mdpi.com/2076-2615/12/19/2706).

Comment 15: Please clarify specifically for what RevMan and Stata were used.

Comment 16: Tables and Figures should stand alone. Therefore, Include in all tables and figures captions to define every abbreviation (e.g., GDM, GL). 

Comment 17: In all tables and all figures, please standardize the style for the references (lowercase/uppercase letters; only the last letter of the authors and year).

Comment 18: Please clarify in Figure 2 if the percentage (%) in the x-axis is the percentage of studies. If so, please include those details in the graphic or a caption.

Comment 19: Table 2 should include a test's P values to determine the heterogeneity's statistical significance. And it should be explained in Materials and Methods how it was determined. 

Comment 20: In table 2, use "[ ] " also in the heading of the column after "Hedges' g," rather than "( )."

Comment 21: Please find a better way to design Table 3. It may be too wide to contain all values in a proper format.

Comment 22: Please explain why 12 g/d was considered the breaking point.

Comment 23: Please detail the procedures used to assess publication bias or refer to them under Materials and Methods.

Comment 24: Under Discussion, please add a note (to work as a disclaimer) stating that funnel plot tests with such a low number of studies may not be conclusive.

Comment 25: In both the abstract and the Discussion, clarify with what the levels >12 g/d were compared. e.g., in lines 246-248, the clause "than..." is missing after the clause "was more effective." (more -> than).

Comment 26: Line 289 refers to "limitations" in the review. Please explicitly add them under a subtitle (non-bolded) in the Discussion.  

Author Response

The authors have conducted an interesting research on the effects of dietary fiber on gestational diabetes. Please consider the following comments:

Thank you for your valuable suggestions. According to your suggestions, I have made the following modifications one by one:

Comment 1: When providing the itemized response to these comments, please ensure the number of the lines where the modifications are located in the SUBMITTED REVISED version of your manuscript. Otherwise, the responses may not be assessed. Thank you.

R1: All revisions werelocated in the submitted manuscript and were shown in the revised format in different colors.

Comment 2: Line 22 - Please clarify what "improving" means.

R2: “improving” on line 22 actually means “reducing”, and the following expressions have been modified. (Line 22)

Comment 3: Authors should explicitly indicate if the meta-analysis is a new one (no other meta-analysis is available on the same topic) or if it is an update. If this is an update, the Discussion should be elaborated accordingly.   

R3: As presented in the introduction of this article, there are a lot of previous articles have shown the beneficial effects of high fiber foods (low GI diet) in pregnant women gestational diabetes blood sugar control. However, no article has described the effect of additional dietary fiber supplement fortification on blood glucose, so the focus of this article is to discuss the effect of dietary fiber supplement fortification on blood glucose, and the comparison of the amount and form of fiber. (Line 68-75)

Comment 4: Please ensure that all materials and methods section is written in the past tense.

R4: All materials and methods were modified to the past tense.

Comment 5: Please clarify criterion #2 in lines 86-87.

R5: Fasting plasma glucose and Blood glucose two hours after meal were the primary outcomes, so at least one had to appear in the included articles. (Line 91)

Comment 6: Please clarify what "people" means in line 90 (human subjects?)

R6: Incorrect statement, I've changed “people” to “human subjects”. (Line 97)

Comment 7: Stating exclusion criterion #3 is redundant with the inclusion criteria and should be removed.

R7: Exclusion criterion#3 has been removed. (Line 97)

Comment 8: Line 94 should say "blinding method."

R8: “blinding method” has been modified. (Line 101)

Comment 9: In lines 94-95, please clarify what "intervention METHOD" and "TYPE" means.

R9: actually the food (dietary fiber) type of intervention. (Line 101)

Comment 10: Line 96. "two were excluded" Two what? Studies?

R10: Has been revised to "two articles". (Line 102)

Comment 11: In line 97, please clarify what "incomplete" means.

R11: Have explained the meaning of “incomplete”. (Line103)

Comment 12: In  line 97, should it be "by email did not respond."

R12: Havemodified the expression. (Line 103)

Comment 13: Please clearly detail the procedure in lines 98-101.

R13: The processes of data extraction and quality assessment were re-described. (Line 101-111)

Comment 14: This is a major observation: In lines 104-105, the value of I2 is used to indicate the significance of the heterogeneity. The terminology "Significance" should be reserved for inferences based on statistical analysis (such as in lines 103-104; P<0.05). However, in lines 104-105, "significance" is used for something different. Please rewrite this sentence and correct it where appropriate across the whole manuscript. Please consider that a meta-analysis with a low I2 could have only trivial heterogeneity but could also have substantial heterogeneity. Conversely, a meta-analysis with a high I2 could have substantial heterogeneity, but could also have only trivial heterogeneity. A concise definition recently used in a partner journal of MDPI is the following: "I2 is defined as the portion (%) of the total variability attributed to pure heterogeneity among studies" (https://www.mdpi.com/2076-2615/12/19/2706).

R14: We have corrected the meaning of I2.( Line 114-117)

Comment 15: Please clarify specifically for what RevMan and Stata were used.

R15: The specifically for Revman and Stata were added. ( Line 113-118)

Comment 16: Tables and Figures should stand alone. Therefore, Include in all tables and figures captions to define every abbreviation (e.g., GDM, GL). 

R16: Definitions have been added for all abbreviations in tables and pictures.

Comment 17: In all tables and all figures, please standardize the style for the references (lowercase/uppercase letters; only the last letter of the authors and year).

R17: The style for the references in all tables and images have been modified.

Comment 18: Please clarify in Figure 2 if the percentage (%) in the x-axis is the percentage of studies. If so, please include those details in the graphic or a caption.

R18: The percent (%) added in figure 2.

Comment 19: Table 2 should include a test's P values to determine the heterogeneity's statistical significance. And it should be explained in Materials and Methods how it was determined. 

R19: P value has been added in table 2.

Comment 20: In table 2, use "[ ] " also in the heading of the column after "Hedges' g," rather than "( )."

R20: The “( )”has been corrected to “[ ]”. (line 198)

Comment 21: Please find a better way to design Table 3. It may be too wide to contain all values in a proper format.

R21: For layout, Table 3 is replaced on a horizontal page.

Comment 22: Please explain why 12 g/d was considered the breaking point.

R22: The 12 g quantity of fiber was determined by the difference between the average dietary intake of 13g fiber9and the recommended intake of 25g to 30g for women in the second trimester of pregnancy. (Line 121)

Comment 23: Please detail the procedures used to assess publication bias or refer to them under Materials and Methods.

R23: The publication bias details have been added in Line 117.

Comment 24: Under Discussion, please add a note (to work as a disclaimer) stating that funnel plot tests with such a low number of studies may not be conclusive.

R24: Added the note about funnel plot.(Line 376-379)

Comment 25: In both the abstract and the Discussion, clarify with what the levels >12 g/d were compared. e.g., in lines 246-248, the clause "than..." is missing after the clause "was more effective." (more -> than).

R25: The level compared with ≥12g/day has been added.

Comment 26: Line 289 refers to "limitations" in the review. Please explicitly add them under a subtitle (non-bolded) in the Discussion.  

R26: The limitation has been added under the discussion.

Round 2

Reviewer 2 Report

Well done.